# The Role of Impella in Cardiogenic Shock in the Post-DanGer Shock Era

**DOI:** 10.3390/biomedicines13092198

**Published:** 2025-09-08

**Authors:** Kassem Farhat, Sara Pollanen, Rongras Damrongwatanasuk, Laura DiChiacchio, Colby Salerno, Nikhil Sikand, Wissam I. Khalife, Jiun-Ruey Hu

**Affiliations:** 1Department of Internal Medicine, Yale School of Medicine, Yale University, New Haven, CT 06510, USA; 2Temerty Faculty of Medicine, University of Toronto, Toronto, ON M5S 3K3, Canada; 3Division of Cardiovascular Medicine, Department of Internal Medicine, University of Louisville, Louisville, KY 40290, USA; 4Department of Cardiac Surgery, Smidt Heart Institute, Cedars-Sinai Medical Center, Los Angeles, CA 90048, USA; 5Department of Cardiology, UMass Chan-Baystate Medical Center, Springfield, MA 01199, USA; 6Section of Cardiovascular Medicine, Yale School of Medicine, Yale University, New Haven, CT 06510, USA; 7Department of Cardiovascular Medicine, The University of Texas Medical Branch, Galveston, TX 77555, USA; 8Department of Cardiology, Smidt Heart Institute, Cedars-Sinai Medical Center, Los Angeles, CA 90048, USA; jiun-ruey.hu@yale.edu

**Keywords:** microaxial flow pump, Impella, cardiogenic shock, myocardial infarction, mechanical circulatory support

## Abstract

The microaxial flow pump (mAFP) is a mechanical circulatory support device designed to directly unload the left ventricle, restore cardiac output, and improve systemic perfusion in the setting of cardiogenic shock (CS). CS is a devastating complication of acute myocardial infarction (AMI) and advanced heart failure, characterized by systemic hypoperfusion and myocardial dysfunction, carrying an in-hospital mortality of 30–50%. However, there has been controversy about whether these theoretical physiological mechanisms behind mAFP translate into actual survival or recovery in this patient population that has historically been difficult to study in prospective trials. The lack of consensus has resulted in differing national guidelines, resource allocation, and clinical decision-making in time-sensitive clinical scenarios. Earlier studies were limited to retrospective analyses and a single small, underpowered randomized trial, none of which showed a mortality benefit. In 2024, the DanGer Shock trial emerged as the first multi-center trial to demonstrate mortality benefit in patients with STEMI complicated by CS treated with Impella CP, albeit at the cost of increased risk of major bleeding, hemolysis, and vascular complications, an effect sustained in the 10 year outcomes published in 2025. In this review, we examine reasons for the differing results of preceding studies and compare how multinational guidelines have reacted to this new evidence. Finally, we provide practical considerations regarding the use, complications, and troubleshooting of this technology, and identify gaps in evidence regarding patient selection and timing of placement.

## 1. Introduction

Cardiogenic shock (CS) is a life-threatening complication of cardiac dysfunction, characterized by inadequate cardiac output (CO) to meet systemic metabolic demands, resulting in hypoperfusion and end-organ injury [1]. Although the etiology of CS can be multifactorial, acute myocardial infarction (AMI) remains one of the leading causes, accounting for approximately 30–40% of cases [2,3]. The high in-hospital mortality associated with AMI-related CS (AMI-CS), approaching 50%, has prompted substantial interest in identifying therapeutic strategies to improve outcomes [1,4,5]. Landmark clinical trials, including SHOCK and CULPRIT-SHOCK, have demonstrated the benefit of emergent revascularization strategies compared to medical therapy alone in patients with AMI-CS, particularly emphasizing immediate coronary angiography with culprit-lesion-only percutaneous coronary intervention (PCI) [5,6,7,8]. Alongside revascularization strategies, extensive effort has been devoted towards mechanical circulatory support strategies, including the intra-aortic balloon pump (IABP), extracorporeal membrane oxygenation (ECMO), and microaxial flow pump (mAFP), which will be the focus of this review.

Over the last decade, the clinical application of the mAFP has grown significantly despite the lack of randomized clinical data supporting its use, giving rise to controversy about its indications and reimbursement [9,10,11]. It was not until the DanGer Shock trial that a significant mortality reduction was demonstrated with mAFP use in STEMI-CS [12]. In this paper, we provide an overview of mAFP’s mechanism of action, explore potential reasons underlying the differing results of earlier studies, and compare how various international guidelines have responded to the evolving body of evidence.

## 2. Technical Specifications and Mechanism of Action

The mAFP is an intravascular, left ventricular assist device (LVAD) that provides continuous axial flow, and which can be inserted percutaneously [13,14]. The system comprises three main components: (1) an internal catheter housing the impeller pump, (2) a drive line connecting the catheter to the external controller, and (3) an external controller that provides power and modulates flow [15]. All mAFP devices are positioned across the aortic valve (AV), with the inlet port inserted in the left ventricular (LV) and the outlet port directed into the ascending aorta. LV unloading is achieved through an impeller, a miniature rotating screw housed within the catheter shaft at the level of the aortic root, that spins at a high-speed generating a pressure gradient drawing blood continuously from the LV to the proximal aorta [14].

### 2.1. Effect on the Pressure Volume Loop

In the setting of AMI-CS, impaired myocardial contractility leads to a low CO state, which in turn elevates LV filling pressures and precipitates both systemic and coronary hypoperfusion [6]. The ensuing hypoperfusion provokes reflex sympathetic activation and compensatory systemic vasoconstriction, which ultimately lead to increased afterload and subsequently myocardial workload and oxygen demand [16]. On the pressure-volume loop, these changes are reflected by an increase in left ventricular end-diastolic volume (EDV) and end-diastolic pressure (EDP) (Figure 1). These pathophysiological changes extend to include neurohormonal and inflammatory responses perpetuating a downward spiral of hemodynamic compromise. Systemic hypoperfusion activates the renin–angiotensin–aldosterone system (RAAS), promoting sodium and water reabsorption, thereby worsening volume overload and ventricular strain [17]. Meanwhile, tissue ischemia and necrosis trigger a vicious inflammatory cycle including increased nitric oxide production, which induces peripheral vasodilation and further worsening hypotension [1,18].

Support from a mAFP provides continuous ventricular unloading throughout the cardiac cycle, reducing left ventricular EDV and EDP (Figure 1). This results in lowering ventricular wall stress and myocardial workload, as well as myocardial oxygen consumption (MVo2) [19,20]. Simultaneously, outflow into the aorta increases CO and mean arterial pressure (MAP), which enhances systemic and coronary perfusion, thereby attenuating ongoing end-organ damage and myocardial injury [6]. In the pressure-volume (PV) loop, use of a mAFP will cause a progressive loss of the isovolumetric contraction and relaxation phases, converting the loop into a triangular shape with a shift leftward and downward [6,19]. Figure 1 illustrates the pathophysiologic alterations in PV loops during CS, and the distinct hemodynamic profiles generated by various MCS devices including mAFP, IABP, and VA-ECMO. Notably, these alterations are directly related to the level of mechanical support, whether partial or full, with greater unloading resulting in a decreased pressure–volume area (PVA), a surrogate marker of myocardial energy expenditure and MVo_2_ [20,21].

### 2.2. Variation in Impella Models

Several models of mAFP are currently available, with subtle differences in function, cannula diameter, insertion techniques, and flow capacity (Table 1). Earlier-generation devices include the Impella 2.5 and CP (Abiomed Inc., Danvers, MA, USA) with 12F and 14F pump diameters, providing up to 2.5 L/min and 3.7 L/min of cardiac output, respectively [13]. The Impella 2.5 has generally been phased out of clinical practice due to limited hemodynamic support, while the CP remains the first option in CS unless prolonged or higher-flow support is needed, in which case escalation is appropriate [19,22]. Both the 2.5 and CP are typically inserted percutaneously via the femoral artery and advanced in a retrograde fashion up the descending aorta, while the axillary and subclavian percutaneous vascular access options are less commonly used due to the risk of upper extremity ischemia [23]. When peripheral vascular anatomy is unfavorable for percutaneous insertion, transaxillary cut-down serves as an alternative access route [23]. In contrast, the newest models of Impella (5.0, left direct [LD], and 5.5; Abiomed Inc., Danvers, MA, USA) are 21F pumps and provide higher flow rates of up to 5.0–5.5 L/min. These devices provide extended support duration and enhanced hemodynamics. They are mostly implanted surgically via axillary artery cut-down or through direct ascending aortic cannulation in open sternotomy cases, whereas the femoral approach is rarely utilized [13]. On the other hand, Impella RP is the only FDA-approved mAFP indicated in the management of right heart failure (HF), providing up to 4.4 L/min of flow. Several structural features should be noted about the Impella 5.5 compared to its predecessors, including the elimination of the flexible pigtail, mitigating the risk of thrombus formation and the disruption of the chordae tendineae of the mitral valve, as well as the introduction of a more rigid cannula and shorter motor segment designed to improve catheter maneuverability [24,25,26]. A side-by-side visual comparison of the most frequently used Impella devices (CP, 5.5, and RP), highlighting their anatomic configurations and technical specifications, is illustrated in Figure 2.

## 3. Outcomes

### 3.1. History

The framework of a mAFP originated in 1985 with the development of the Hemopump by Dr. Richard Wampler, although it was never brought to market [27]. Building on this foundation, in the late 1990s, Dr. Thorsten Siess conceptualized a catheter-based mAFP designed to actively unload blood from the LV into the ascending aorta [28]. This innovation led to the development of the mAFP device, which was later commercialized as Impella and acquired by Abiomed in 2005. Around the same time in Europe, experimental work led by Meyns and colleagues supported the clinical approval of mAFP, demonstrating a reduction in infarct size in animal models of AMI when mAFP support was initiated prior to revascularization [29,30].

The first introduction into the US market was in 2008, when Impella 2.5 received the U.S. Food and Drug Administration (FDA) 510(k) clearance for use as a temporary MCS device during high-risk PCI (HR-PCI) [31]. This was followed by multiple FDA clearances and pre-market approvals (PMA) over the last decade, expanding mAFP’s clinical indications. A detailed visualization of the timeline for the introduction of various mAFP devices into the market is presented in Figure 3.

### 3.2. Observational Studies

Over the past decade, mAFP use has expanded substantially, despite limited high quality evidence for its efficacy [32,33]. Early observational data were encouraging, notably from the USpella registry, and the multicenter RECOVER I trial, demonstrating favorable outcomes and contributing to mAFP’s PMA approval [11,34,35]. In the USpella registry, early Impella 2.5 use during HRPCI was associated with improved survival versus post-PCI placement (65.1% vs. 40.7%; *p* = 0.003) [34]. The RECOVER I trial also showed favorable outcomes with the use of Impella 5.0/LD in cardiac surgery-related CS, with a 13% composite endpoint of stroke or death at 30 days [11].

Building on this foundation, the Detroit Cardiogenic Shock Initiative, a pilot feasibility study, evaluated pre-PCI mAFP support in AMI-CS, demonstrating improved survival to 85% compared to 51% in historical controls (*p* < 0.001) [36]. This laid the groundwork for the National Cardiogenic Shock Initiative (NCSI), a prospective multicenter study of 171 patients with a 72% survival rate at discharge using early MCS in a structured protocol [37]. The cVAD registry showed higher survival with early vs. delayed mAFP use (46% vs. 35%; OR 0.485; *p* = 0.04), and a Japanese registry analysis reported an 81% 30-day survival in AMI-CS patients receiving mAFP support [38,39].

The aforementioned highlight the ambiguity surrounding the clinical utility of mAFP in AMI-CS. Beyond limited RCT evidence, several studies did not only fail to show mortality benefits with mAFP but have also raised concerns over higher complication rates [31,32,33,40,41,42,43]. For instance, a matched-pair analysis by Schrage et al. compared 237 patients receiving mAFP support with IABP-treated patients from the IABP-SHOCK II trial and found no mortality benefit but significantly more life-threatening bleeding (8.5% vs. 3.0%, *p* < 0.01) and vascular complications (9.8% vs. 3.8%, *p* = 0.01) [44]. Dhruva et al., in a large retrospective cohort (n = 3360), reported increased mortality (45.0% vs. 34.1%) and major bleeding (31.3% vs. 16.0%) with mAFP versus IABP [45]. Consequently, utilization patterns vary widely across hospitals and are primarily driven more by expert opinion than robust randomized data [32,46,47].

### 3.3. Randomized Trials

However, randomized trials of mAFP support failed to show promising results but were drastically underpowered (Table 2). In 2008, ISAR-SHOCK (n = 25) demonstrated superior hemodynamics with Impella 2.5 vs. IABP, significantly increasing cardiac index at only 30 min (0.49 vs. 0.11 L/min/m^2^; *p* = 0.02), but there was no difference in 30-day mortality (46% for both) [9]. In 2012, the first large RCT (PROTECT II, n = 448) to compare Impella 2.5 with IABP in HR-PCI, showed similar 30-day major adverse cardiac events (MACE), though stroke was lower in the mAFP group (0% vs. 1.8%, *p* = 0.04) [48]. Although terminated for futility, there was a trend toward lower 90-day events with mAFP in both intention-to-treat (40.6% versus 49.3%, *p* = 0.066) and per protocol analysis (40.0% versus 51.0%, *p* = 0.023) [48]. The IMPRESS trial (n = 48), which randomized AMI-CS patients in a 1:1 fashion to Impella CP or IABP, did not show a mortality difference. The overall 6-month mortality exceeded 50%, largely attributed to the critical illness of the enrolled cohort, including neurological damage post cardiac arrest [10]. Of note, the IMPELLA-STIC, a small RCT on AMI-CS, where patients already on IABP received added mAFP support, reported no hemodynamic benefit but higher complications [49].

Proponents of mAFP argued that these trials were underpowered and enrolled severely ill patients masking any potential survival advantage [9,10,48,50]. These studies, together with evolving clinical experience, have shaped the regulatory and investigational trajectory of Impella devices over the past decade. Figure 3 provides a chronological view of key approvals and pivotal trials, situating DanGer Shock within the broader landscape of Impella development.

Beyond the limited supporting clinical data, several additional factors influence the widespread adoption of mAFP devices, including healthcare regulatory frameworks, reimbursement systems, and hospitalization costs [31,51,52]. Notably, medical device approval and reimbursement policies are more favorable in the United States compared with more restrictive approaches in Canada and Europe [52]. The costs of mAFP use exceed those of patients who do not receive MCS, reflecting not only the direct expense of the device itself but also the additional costs related to concomitant interventions and post-procedural care [53]. In a database study of 2722 insured patients with HF-CS between 2010 and 2019 in the United States, there was not only a significant increase in the rate of mAFP use (+344%), but also a linear association between higher cost quartiles and complication rates. Importantly, adjusted hospitalization costs for mAFP were greater than for IABP (median USD 142,518 [IQR, USD 126,845–USD 179,938] vs. USD 132,060 [IQR, USD 113,794–USD 160,244]), though lower than for ECMO (median USD 191,079 [IQR, USD 165,760–USD 239,373]) [54]. Evidence regarding the cost-effectiveness of Impella is mixed and remains controversial, with industry-sponsored studies often reporting favorable outcomes, while independent evaluations generally show higher costs without clear benefit [55]. Updated real-world studies from the past decade are needed to better assess its cost-effectiveness and clinical impact.

### 3.4. mAFP in the DanGer Shock Era

In 2024, the DanGer Shock trial ushered a pivotal shift, providing the first randomized evidence supporting early mAFP use in the AMI-CS population [12]. This international, multicenter RCT, conducted across Germany, Denmark, and the United Kingdom, enrolled 355 patients who were randomized to receive Impella CP for at least 48 h (n = 179) or standard of care (n = 176). At 180 days, a 12.7% absolute risk reduction in mortality was observed (45.8% vs. 58.5%, *p* = 0.04), [12]. However, these effects were counterbalanced by a significant higher complications rate in the intervention group (24.0% vs. 6.2%), including moderate or severe bleeding (RR: 2.06, 95% CI: 1.15–3.66) and vascular complications (RR: 5.15, 95% CI: 1.11–23.84), renal replacement therapy (RR: 1.98, 95% CI: 1.27-3.09), and sepsis (RR: 2.79, 95% CI: 1.20–6.48) [12]. While the trial took nearly a decade to overcome recruitment and logistical barriers, it represents the first positive randomized trial of a device in AMI-CS and marks an important milestone in the field.

Notably, only ST elevation myocardial infarction (STEMI) patients were enrolled, as previously performed solely in the IMPRESS trial. Conversely, high-risk subsets, including those with out-of-hospital cardiac arrest (OHCA) complicated with coma (Glascow coma scale score < 8), right heart failure (RHF), or shock lasting >24 h were excluded. Accordingly, only ~20% of DanGer Shock patients had cardiac arrest compared to >90% in IMPRESS, whereas among prior trials, RHF exclusion was only seen in the ISAR shock trial. This selective approach led to a low inclusion rate (~30%; 360 of 1200 screened), shaping the trial into a targeted intervention for a narrowly defined population. A recent registry analysis from the Critical Care Cardiology Trials Network (CCCTN) similarly found that only 5% of CICU patients with cardiogenic shock and 32% with STEMI-CS would meet DanGer Shock eligibility criteria for mAFP support [56].

Regarding shock protocols, the trial employed a rigorous and timely system, distinguishing it from prior trials. Over 80% of patients received mAFP at the catheterization laboratory, 56.6% pre-PCI and 27% post-PCI [12]. By contrast, only 20% of patients in the IMPRESS trial received it before PCI, while ISAR-SHOCK patients received the device only post-revascularization [9,10]. In this context, a large retrospective study from the IQ registry (n = 15,259) demonstrated a better survival rate in pre-PCI patients compared to salvage therapy (56% vs. 52%; *p* < 0.001) [57]. The DanGer Shock trial also incorporated pulmonary artery catheters (PAC) use in a substantial proportion of enrolled patients (68%) to guide escalation and de-escalation of therapy, aligning with the contemporary SCAI SHOCK framework that implements a structured, stage-based decision-making guided by serial hemodynamic measurements [12,58,59,60]. In contrast, earlier trials lacked routine PAC integration into treatment plans, limiting the precision in patient selection and therapeutic escalation [59,60].

After extending follow-up to six months, surpassing the 30-day endpoints used in earlier studies, DanGer Shock further showed that at 10 years, there was a sustained reduction in all-cause mortality in patients in the Impella arm compared to standard care. Given that center experience can influence clinical outcomes, the conduct of the DanGer Shock trial at well-established high-volume European centers, with experienced personnel and advanced infrastructure capable of implementing complex shock algorithms effectively, may have contributed to its findings [12,57]. Similar observations came out from Japan showing an 81% 30-day survival, likely enhanced by restricting mAFP use to qualified facilities with trained operators [61]. Although definitive evidence is lacking, improved outcomes have been correlated with institutional specialized teams and higher case volume across CS, HR-PCI, and MCS contexts [57,62,63,64]. Yet, despite its limitations, the DanGer Shock trial has influenced guideline updates.

## 4. Guidelines and Indications

The results of the DanGer Shock trial influenced the recommendations of several cardiology societies (Table 3). The 2021 European Society of Cardiology (ESC) heart failure guidelines were the first to elevate mAFP’s status, upgrading their recommendation from Class IIb to Class IIa for use in AMI-CS, even before the completion of the DanGer Shock trial [65]. In practical terms, the growing body of evidence supports the use of mAFP as a reasonable intervention to reduce mortality in patients with STEMI-CS. This was concurrently applied with downgrading the IABP to Class III (“not recommended”) [65].

More recently, the 2025 ACC/AHA guidelines have also upgraded their recommendation for mAFP use in AMI-CS from Class IIb to Class IIa, with a Level of Evidence B-R [66]. However, it is believed that the strength of recommendations may have been higher had the device not been associated with an increased risk of complications, and if greater clarity regarding the optimal timing of deployment existed.

The 2025 clinical guidelines by the National Heart Foundation of Australia acknowledged the emerging data from the DanGer Shock trial but expressed concerns regarding the reported adverse events and the trial’s highly selective population. As a result, the guideline rated the strength of evidence as moderate and issued only a weak recommendation for mAFP use, emphasizing individualized, case-by-case decision-making [12,67]. Similarly, the 2024 Chinese guidelines for the diagnosis and management of HF endorsed mAFP as a reasonable temporary therapeutic option in selected patients, particularly as a bridge to recovery or escalation to advanced therapies, assigning it a Class IIa recommendation [68].

Meanwhile, several regional and national guidelines were issued prior to the publication of DanGer Shock. The 2024 Asian Pacific Society of Cardiology (APSC) consensus statement on the management of AMI-CS endorsed the temporary use of MCS in SCAI Stage C or D patients, while 79% of the panel deemed its use in stage E to be futile [69]. The 2017 Comprehensive Update of the Canadian Cardiovascular Society Guidelines (CCS) for the Management of Heart Failure strongly recommended the use of temporary MCS, including mAFP, in patients with CS pending evaluation for long-term therapeutic options [70]. In contrast, the National Institute for Health and Care Excellence (NICE) of the United Kingdom has not issued national guidance on the use of MCS in CS [71]. In Japan, the 2023 focused updates from the Japanese Circulation Society (JCS) noted favorable 30-day survival outcomes in patients with AMI-CS supported with mAFP from the J-PVAD registry [61,72]. However, these updates preceded the DanGer shock trial results and did not specify a formal class recommendation for mAFP use [72].

In summary, not all international societies have yet incorporated the recent updates from the DanGer Shock trial into their official guidelines, though further revisions are anticipated as society meetings convene.

**Table 3 biomedicines-13-02198-t003:** Impella Recommendations Across Major Cardiology Societies (as of 2025).

Society/Region	Class/Level		Notes
	Cardiogenic Shock	High-Risk PCI	
ACC/AHA (USA) [66]	Class IIa/Level B-R	Class IIb/Level B-R	“In selected * patients with STEMI and severe or refractory cardiogenic shock, insertion of a microaxial intravascular flow pump is reasonable to reduce death.”
ESC (Europe) [73]	Class IIa/Level C	Class IIb/Level C	“Short-term MCS should be considered in patients with cardiogenic shock as a BTR, BTD, BTB. Further indications include treatment of the cause of cardiogenic shock or long-term MCS or transplantation.”
NICE (UK) [71]	No formal class; cautious use in expert centers	No formal class; supported in specialized centers	“No national guidance exists in the UK for the use of hemodynamic support devices.”
CCS (Canada) [70]	Strong Recommendation; Moderate-Quality Evidence	Not formally recommended outside research	“We recommend that patients in cardiogenic shock be considered for temporary MCS to afford an opportunity for evaluation for long-term options.”
JCS (Japan) [72]	N/A	N/A	“However, all of these data are from clinical trials with small sample sizes, there is a lack of high-quality RCTs, and interestingly, to date there are no specific recommendations in guidelines from Europe and North America, where IMPELLA is more widely used than in Japan.”
CSC (China) [68]	Class IIa/Level B	No formal class	“Percutaneous ventricular assist devices and extracorporeal membrane oxygenation: These devices can be utilized as transitional therapies for fulminant myocarditis, acute severe HF, or cardiogenic shock, allowing for further evaluation of the need for heart transplantation or long-term MCS (IIa, B).”
ANZ (Australia/ New Zealand) [67]	Strength of recommendation weak with moderate certainty of evidence	N/A	“Consider left ventricular assist devices in people with STEMI and cardiogenic shock on a case-by-case basis, given the selected population enrolled and the complication rate in the DanGer Shock trial.”
Asia-Pacific Region [69]	Low level of evidence	N/A	“Temporary MCS (e.g., intra-aortic balloon pump [IABP], Impella or venoarterial extracorporeal membrane oxygenation [VAECMO]) may be considered in AMI patients in Stage C and Stage D CS.”

* Data are based on the findings of the DanGer Shock trial that included a selective population of patients with AMI-CS. NICE: National Institute for Health and Care Excellence. CCS: Canadian Cardiovascular Society. CSC: Chinese Society of Cardiology. N/A: Information not available in current society guidelines.

## 5. Troubleshooting and Complications

Optimal deployment of mAFP requires technical proficiency and close post-implantation monitoring to detect and manage complications. These adverse events primarily arise from vascular access issues or catheter malposition.

### 5.1. Systematic Approach to Interpreting Impella Console Alarms and Waveforms

Before analyzing mAFP console waveforms, a focused bedside evaluation, including auscultation of the characteristic humming sound of the continuous-flow device, as well as insertion site and external circuit inspection, is imperative. Three essential distinct waveforms are displayed on the AIC: the aortic (red) waveform, the LV (white) waveform, and the motor current (green) waveform (Figure 4).

The red pressure waveform is a placement signal measured by an optical sensor located at the junction of the cannula and outflow and is useful for confirming the proper positioning of the catheter in the ascending aorta. This waveform should demonstrate pulsatility and resemble a typical arterial pressure tracing. The white waveform is an estimated LV placement signal derived from the aortic waveform and the pressure gradient between the left ventricle (inlet) and aorta (outlet). The pressure gradient is calculated based on motor speed and current fluctuations during the different phases of the cardiac cycle. While the green motor current waveform reflects resistance to flow through the device and should remain pulsatile and at stable values relative to the performance level (P-level).

Furthermore, position alarms typically indicate catheter migration, while suction alarms generally result from low preload, malposition, or right ventricular (RV) failure. The differential diagnosis of a suction alarm depends on the accompanying waveform pattern and hemodynamic parameters such as central venous pressure (CVP) or right atrial (RA) pressure, which can help prioritize potential causes. Purge pressure alarms reflect abnormalities within the purge system; low pressures may suggest leaks or dilute purge solution, whereas high pressures can indicate line kinks or thrombus formation. Lastly, a persistently elevated motor current should raise concern for increased impeller load or thrombus within the device. More indicators and parameters supporting mAFP’s function monitoring are represented in Figure 4.

### 5.2. Malposition

Appropriate positioning is essential to ensure LV unloading and avoid complications. A Multimodality imaging approach, incorporating both fluoroscopy and echocardiography, is employed to confirm proper positioning [74,75]. Applying the Doppler feature during echocardiography offers additional insight into mAFP positioning. In proper deployment, a dense mosaic pattern of flow turbulence is typically seen just above the aortic valve, near the catheter outlet. However, the presence of turbulence on both sides of the valve may suggest that the device is inserted too deeply. Malpositioning can lead to abrupt hemodynamic deterioration and systemic hypoperfusion, often triggering machine alarms [76].

For optimal placement, it is essential to pay attention to key position parameters that encompass the depth of the catheter across the AV and its rotational alignment within the LV. While manufacturers provide procedural manuals to guide insertion and repositioning, these instructions may not always account for patient-to-patient anatomical variations [74].

Also, the catheter inlet should be positioned 3.5 cm below the AV (5 cm for the 5.5 model), the outlet above the AV, and the catheter tip directed at the LV apex (Figure 5) [74,75]. Proper positioning can be assessed using the parasternal long-axis (PLAX) view on transthoracic echocardiography (TTE), as shown in Figure 5, where key anatomic landmarks and components of the mAFP are clearly visualized.

#### 5.2.1. Deep Malposition

When advanced too deeply such as the inlet lodged between papillary muscles and the myocardial wall, the outlet port may become partially or entirely beneath the AV resulting in diminished forward aortic blood flow [77]. Console signs may include loss of clear differentiation between aortic and ventricular pressure waveforms, with the aortic tracing beginning to resemble the LV waveform, suggesting that the outlet may no longer be positioned within the aorta. Additionally, a ventricular placement signal waveform may appear, and flow disturbances may trigger “suction” alarms [78].

#### 5.2.2. Shallow Malposition

Shallow malposition occurs when the inlet sits too close to the AV, often indicated by a flat motor current and a pulsatile aortic pressure waveform on the console. This typically triggers the “Impella in the aorta” alarm. On echocardiography, the pigtail may appear close to the mitral valve, with the inlet barely within the LV cavity or even straddling the AV, resulting in suboptimal unloading and ineffective systemic perfusion [78]. Figure 6 provides a PLAX echocardiographic view illustrating the malpositioning, along with recommended strategies for adjustments.

#### 5.2.3. Malrotation

Although often underrecognized and primarily detected via echocardiography, malrotation is relatively common, occurring in approximately one-third of cases [76]. It refers to deviation of the inlet tip away from the LV apex toward the mitral valve apparatus or inferolateral wall despite appropriate depth and unremarkable console tracings [76]. While hemodynamic parameters may remain unchanged, malrotation is often associated with suboptimal unloading of the LV.

#### 5.2.4. Correcting Malposition

While fluoroscopy is the most common modality for device insertion, echocardiography remains the first-line tool to confirm positioning [77,79]. Once malposition is suspected, echocardiography allows direct visualization of both the inlet and outlet ports, guiding safe and appropriate repositioning by advancing or retracting the catheter. Prior to manipulation, support should be reduced to P-2 to prevent suction or myocardial injury, the Tuohy-Borst (TB) valve unlocked to allow 1 cm incremental adjustments, with live echocardiographic guidance, continuous monitoring of console waveforms, and hemodynamics [78]. Given potential slack in the system, it is important to wait a short period of time to assess delayed transmission of the adjustment before making another 1 cm adjustment. Once optimal positioning is restored, the valve is locked, and support gradually increased [78]. If the device is completely dislodged into the aorta, repositioning across the AV should not be attempted without a guidewire, due to the risk of valve or vascular injury.

Malrotation correction is more complex due to the catheter lacking inherent torque transmission within the LV cavity, requiring gentle clockwise rotation under echocardiographic guidance to redirect the inlet tip toward the LV apex [78]. For difficult cases or poor echo windows, fluoroscopic-guided repositioning may be necessary. If unsuccessful, the device should be removed and exchanged [78].

### 5.3. Thrombosis

Thrombus formation may impair flow leading to suboptimal hemodynamic support [78]. Though the incidence is not fully defined, it increases with prolonged use and is closely associated with hemolysis, which has been reported in 30% to 60% of patients, depending on the study and definition used [13,80,81].

Thrombus may form at the inlet, cannula, or outlet, and is often reflected in altered pump metrics, including high purge pressure, flat motor waveforms and an abnormal aortic placement signal, which may mimic catheter malposition. The system typically generates console alarms such as “suction” and “Impella position unknown”, prompting further evaluation. In addition, the high shear stress across obstructed structures contributes to red cell destruction and hemolysis [81].

Following insertion, all patients receive systemic anticoagulation with intravenous unfractionated heparin (UFH) and a heparinized dextrose purge solution continuously infused through the motor housing to create a pressure barrier against blood inflow, reducing the risk of thrombosis and maintaining catheter patency [82]. Although the standard purge solution traditionally consists of dextrose for viscosity and heparin for anticoagulation, sodium bicarbonate-based purge solutions have been widely adopted as an alternative to heparin, not only in cases of contraindication but also as a preferred option across many centers [82].

Suspected thrombosis should prompt a hemolysis workup, echocardiography to exclude device malposition or an underfilled LV, and confirmation of therapeutic anticoagulation [activated clotting time (ACT) 160–180 s]. If ACT remains subtherapeutic, UFH should be escalated, or non-heparin-based alternatives (e.g., bivalirudin or argatroban) should be used if resistance or heparin-induced thrombocytopenia (HIT) is suspected [83]. In refractory cases with persistent obstruction or significant thrombus burden, device exchange or explantation should be considered [78].

### 5.4. Hemolysis

Hemolysis is another major complication, more commonly encountered in mAFP than IABP, with reported incidence ranging between 5% and 63%, warranting close monitoring [81,84,85]. It is primarily driven by shear stress, especially at high impeller rotational speeds, and is exacerbated by device thrombosis or malpositioning [86]. Hemolysis can trigger a prothrombotic cascade via the release of free hemoglobin, which scavenges nitric oxide, leading to increased vascular tone and platelet activation [81,87]. It may also precipitate acute kidney injury [81,87].

Clinically, hemolysis may appear as dark-colored urine, anemia or jaundice. Daily laboratory surveillance is essential, and suggestive findings include elevated plasma-free hemoglobin, increased lactate, lactate dehydrogenase (LDH) levels > 2.5 times the upper limit of normal, and unconjugated bilirubin, typically accompanied by a decline in haptoglobin [81,82].

Once identified, evaluation should first focus on device-related causes such malposition and thrombosis, inadequate LV preload, RV dysfunction, then underlying hematological disorders when initial workup is inconclusive [81].

### 5.5. Vascular Access

Given the large sheath size and complexity of mAFP insertion, meticulous vascular access management is essential to mitigate potentially fatal complications such as severe bleeding and limb ischemia [88]. mAFP use is associated with more access complications than IABP, though less than ECMO [44,89].

Reported bleeding rates vary across studies due to inconsistent definitions [90]. Causes of access-site bleeding include introducer-sheath mismatch, large catheter size, and suboptimal insertion techniques [13]. Manual compression is the most cost-effective approach for managing bleeding. Following device removal, femoral compression systems or vascular closure devices can be employed to reduce bleeding risk, while surgical removal may be required in complex cases [19].

Furthermore, limb ischemia is a serious complication of lower limb hypoperfusion that often necessitates intervention [91]. It occurs primarily due to small-caliber femoral arteries observed mostly in women, as well as vasopressor use, and preexisting peripheral arterial disease [88,92]. A meta-analysis of both RCTs and observational studies (n = 2121) reported a 6.1% incidence of limb ischemia in patients supported with mAFP [93]. Early detection relies on clinical examination, with Doppler ultrasound as a helpful adjunct, though angiography remains the diagnostic gold standard [94]. Given the urgency of this complication, preventive strategies, such as distal perfusion catheter placement, can be considered. Therapeutic interventions may include femoral–femoral bypass, open thrombectomy, or limb amputation in severe cases [95,96]. The Impella catheter should be inserted into the femoral artery at a 30–45° angle, with use of gauze above the skin to maintain this trajectory. Attention should be given to ensure that this angle of insertion remains throughout the patient’s ICU course. It is also important to note that the axillary artery lies closely to the brachial plexus and is enclosed within the same neurovascular sheath alongside the axillary vein and nerves. Therefore, a careful surgical technique is essential to minimize the risk of plexus injury [25].

## 6. Unanswered Questions

Despite the numerous observational and randomized studies, several key questions regarding mAFP use in AMI-CS remain unanswered. The optimal timing of device deployment is a major issue that carries significant clinical implications. While several studies reported improved outcomes associated with early insertion, robust RCT-level evidence remains lacking [38,97,98]. For instance, a 2022 meta-analysis found significantly lower short-term and midterm mortality with pre-PCI mAFP use compared to post-PCI (37.2% vs. 53.6%, 47.9% vs. 73%, respectively) [98]. However, in the DanGer Shock trial, although 84% were randomized in the cath lab, only about half received mAFP before revascularization [12]. This limited application of true early unloading makes it challenging to draw definitive conclusions regarding the impact of LV wall stress reduction via early mechanical unloading on outcomes beyond general hemodynamic stabilization [12].

To address these knowledge gaps, multiple trials are underway. In Europe, the ULYSS trial (NCT037657594) is evaluating the efficacy of pre-PCI Impella CP support in patients with ACS complicated by CS [99]. Its inclusion and exclusion criteria mirror those of the DanGer Shock trial, with a stronger emphasis on early MCS deployment and using 30-day composite outcomes rather than 6 months [99]. The UNLOAD-HF-CS trial (NCT05064202) is evaluating the use of Impella 5.5 in CS secondary to acute decompensated heart failure (ADHF), a population not included in the DanGer Shock trial, while specifically excluding cases of AMI and OHCA [100]. Conversely, in the United States, the RECOVER IV, a planned large RCT of mAFP in AMI-CS, was recently suspended by its Data and Safety Monitoring Board (DSMB) due to a perceived lack of clinical equipoise, following the strikingly positive DanGer Shock findings [101]. This study would have provided valuable insights on the use of mAFP in the U.S. patient population. Further studies are needed to assess the durability of Impella’s benefits and to determine the optimal initiation timing in AMI-CS.

Another unresolved concern is the limited generalizability of DanGer Shock’s findings. Only 30% of screened patients were enrolled, reflecting the study’s focus on a narrow subset of CS phenotypes. Excluded populations included those with prolonged shock (>24 h), AMI-related mechanical complication, and OHCA [12]. These exclusion criteria indicate that the trial’s population does not mirror the full spectrum of real-world AMI-CS. Moreover, only 21% of enrolled patients were women, with no report on racial or ethnic distribution. Conducting the trial required a highly skilled shock team, early recognition of CS, and timely informed consent; factors that may limit reproducibility in real-world settings.

Interestingly, subgroup analyses suggested that male patients and low MAP (<63 mmHg) were associated with greater survival benefit from mAFP support [12]. This supports a potential shift toward a more targeted approach for mAFP in a selected population rather than a universal application. Validation through real-world data is critical to refine patient selection strategies.

Lastly, device-related complications continue to pose a major barrier to guideline upgrade. Although survival weas prioritized in DanGer Shock, the net clinical benefit is uncertain, as the number needed to treat (NNT = 8) exceeded the number needed to harm (NNH = 6) [12]. Mitigating complication rates requires multiple elements to succeed. This includes improved selection criteria, refined implantation techniques, newer device iterations, and standardized post-implant protocols, particularly optimal hemodynamic management and support duration [102,103,104,105].

## 7. Conclusions

mAFP offers hemodynamic support in AMI-CS by unloading the left ventricle and improving perfusion. The DanGer Shock trial provided the first randomized evidence of a mortality benefit, marking a critical shift in the field. However, this progress is hindered by high complication rates and limited generalizability of reported findings. Future trials are needed to confirm these results and address unanswered questions, pending additional real-world data and forthcoming updates from professional societies.

## Figures and Tables

**Figure 1 biomedicines-13-02198-f001:**
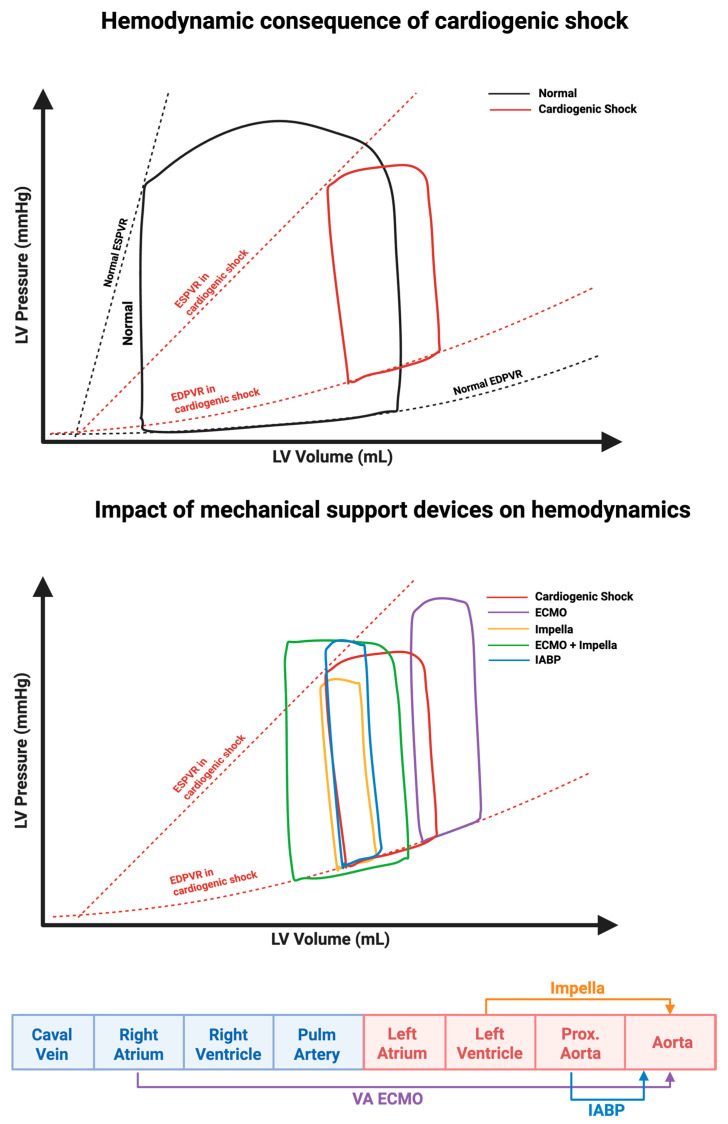
Effects of cardiogenic shock and mechanical circulatory support on LV Pressure–Volume (PV) loops. **Top panel**: Reduced contractility and increased filling pressures seen in the PV loop of cardiogenic shock (red loop) compared to normal physiology (black loop). This results from a rightward shift in the End-Systolic Pressure-Volume Relationship (ESPVR) and an upward shift in the End-Diastolic Pressure-Volume Relationship (EDPVR). **Bottom panel**: Mechanical support devices alter PV loop geometry—Impella unloads the LV, IABP modestly improves hemodynamics, and VA-ECMO increases afterload unless combined with unloading. **Bottom strip**: While the Impella facilitates flow the LV to the aorta, the IABP facilitates flow from the proximal aorta to the descending aorta, and VA-ECMO facilitates flow from the right atrium to the aorta. Original illustration created in BioRender (2025). https://biorender.com/7ehqejy.

**Figure 2 biomedicines-13-02198-f002:**
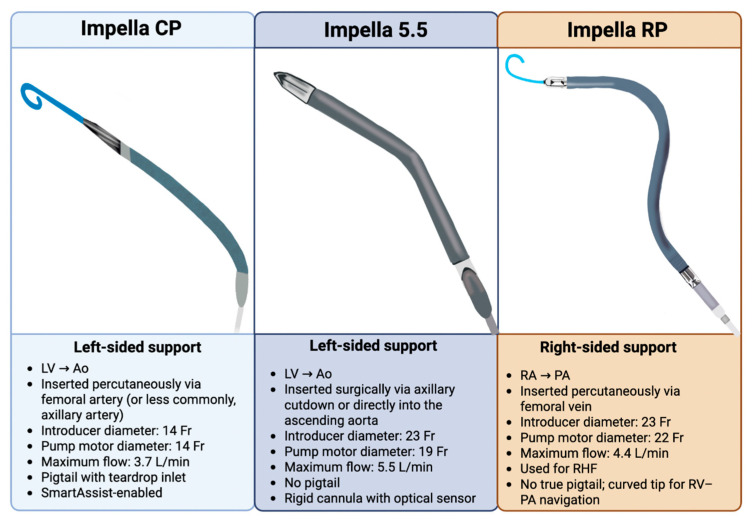
Overview of commonly used Impella devices. Visual comparison of Impella CP, 5.5, and RP. CP and 5.5 provide left-sided support from the LV to the aorta, differing in access method, pump design, and flow capacity. Original illustration created in BioRender. (2025) https://BioRender.com/2z03vvl.

**Figure 3 biomedicines-13-02198-f003:**
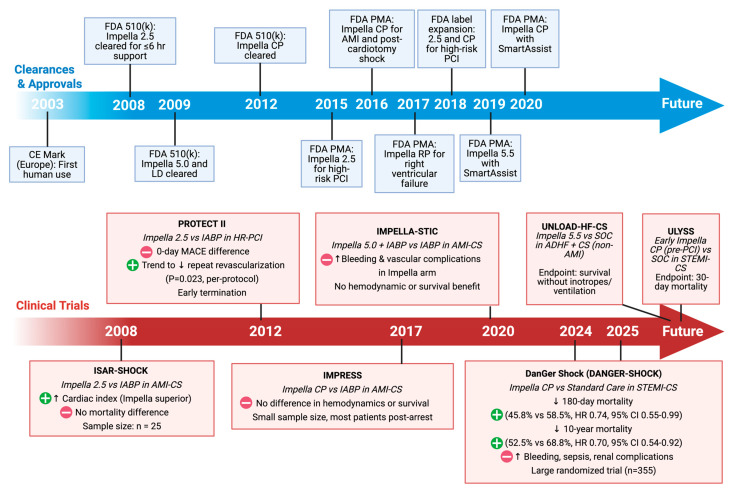
Evolution of Impella device approvals and clearances and major clinical trials. Timeline depicting CE and FDA approvals and clearances (**top**) alongside landmark clinical trials (**bottom**) evaluating mAFP use in high-risk PCI and CS, culminating in the pivotal DanGer Shock trial and future studies. ↓: Decrease; ↑: Increase; (-): Negative outcomes. (+): Positive outcomes. Original illustration created in BioRender. (2025) https://BioRender.com/33pq6rq.

**Figure 4 biomedicines-13-02198-f004:**
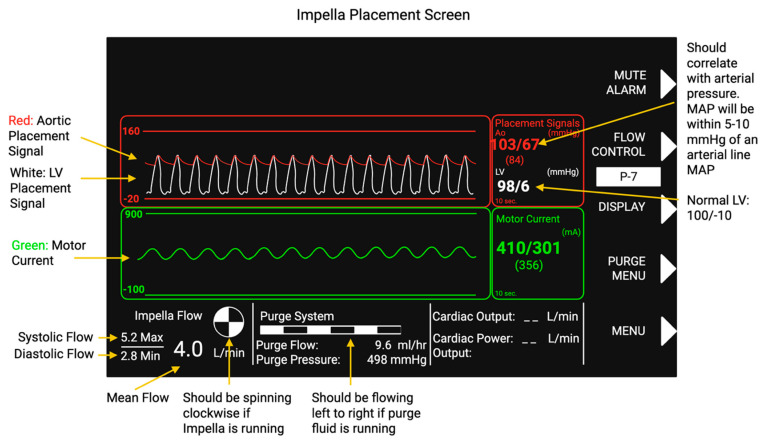
Impella placement screen with pressure and motor waveform tracings. Display panel showing real-time Impella placement signals. Red waveform reflects aortic pressure, white waveform indicates LV pressure, and green waveform represents motor current. Original illustration created in BioRender. (2025) https://BioRender.com/l1fke15.

**Figure 5 biomedicines-13-02198-f005:**
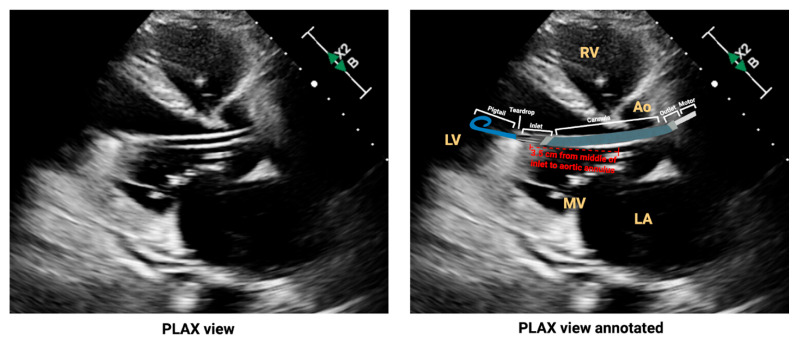
Verifying correct placement of the Impella CP device on transthoracic echocardiography. For reader understanding, this echocardiogram is provided unannotated (**left panel**) and annotated (**right panel**). Abbreviations: Ao = aorta, LV = Left ventricle, LA = left atrium, MV = mitral valve, RV = right ventricle. Original illustration created in BioRender. (2025) https://BioRender.com/d86wclk.

**Figure 6 biomedicines-13-02198-f006:**
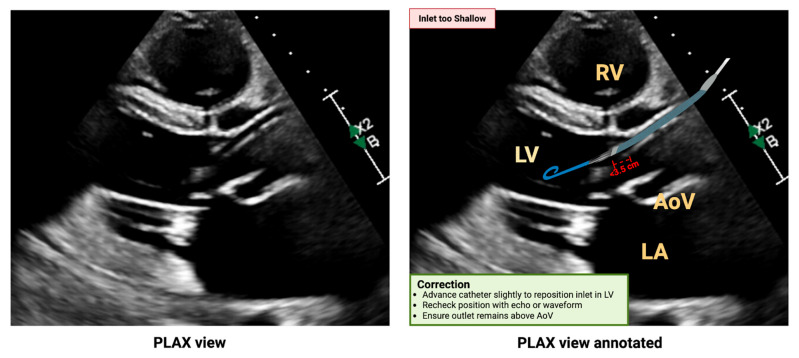
Shallow Impella CP position on echocardiography. PLAX view showing shallow malposition of the Impella with the inlet too close to the aortic valve, unannotated (**left panel**) and annotated (**right panel**). Abbreviations: Ao = aorta, LV = Left ventricle, LA = left atrium, RV = right ventricle. Original illustration created in BioRender. (2025) https://BioRender.com/y43v9a1.

**Table 1 biomedicines-13-02198-t001:** Different Versions of Impella.

	Impella 2.5 (Historical)	Impella CP (Current)	Impella 5.0 (Historical)	Impella LD (Historical)	Impella 5.5 (Current)
Indication	HR-PCI and CS	HR-PCI and CS	CS	CS	CS
Access	Percutaneous femoral or axillary ^1^	Percutaneous femoral or axillary	Femoral or axillary cutdown	Direct insertion into AA	Axillary cutdown or direct insertion into AA
Motor Size (Fr)	12	14	21	21	19
Catheter Size (Fr)	9	9	9	9	9
Sheath Size (Fr)	13	14	23	23	23
Max Flow Rate (L/min)	2.5	3.7	5	5.3	5.5
Max Speed (rpm)	51,000	51,000	33,000	33,000	33,000
Performance Levels	P0–P9	P0–P9	P0–P9	P0–P9	P0–P9
Duration of support	HR-PCI: ≤6 h/ CS: ≤4 d	HR-PCI: ≤6h/ CS: ≤4d	14 days	14 days	14 days
SmartAssist	No	Yes	No	No	Yes
Pressure Sensor	Fluid transducer	Optical sensor	Fluid transducer	Fluid transducer	Optical sensor
Guide Wire	0.018″ diameter × 260 cm placement guidewire			Wireless possible if direct insertion	
Anticoagulation (AC)	Heparin purge + systemic AC or BBPS alone ^2^				
Common Complications	Hemolysis, limb ischemia, bleeding, stroke, infection				
Contraindications (CI)	Severe AS/AR, mechanical AV, LV thrombus, CI to AC				

Abbreviations: AC, anticoagulation; AV, aortic valve; AR, aortic regurgitation; AS, aortic stenosis; CI, contraindication; Fr: French; HR-PCI: high-risk percutaneous coronary intervention, AA: Ascending aorta, BBPS: Bicarbonate-based purge solution. ^1^ Mostly femoral access. ^2^ In case of coagulopathy or persistent significant bleeding despite holding systemic anticoagulation.

**Table 2 biomedicines-13-02198-t002:** Summary of Major Impella Studies.

	ISAR-SHOCK	PROTECT II	IMPRESS	DanGer Shock
Enrollment Period	2004–2007	2007–2010	2012–2015	2013–2023
N (Patients)	25	452	48	355
Population	AMI-CS revascularized by PCI	HR-PCI in patients with reduced LVEF	STEMI-CS revascularized by PCI	STEMI-CS revascularized by PCI or CABG
Exclusion Criteria	-Age < 18 -Valvular disease (MV; severe AR) -Resuscitation > 30 min -CS from AMI-related mechanical complications -RV failure -Other conditions (HOCM, LV thrombus, cerebral disease, PE, sepsis, coagulopathy, bleeding requiring surgery, pregnancy)	-Recent MI with persistent elevated cardiac enzymes -LV thrombus -Platelet count ≤ 75,000/mm^3^ -Creatinine ≥ 4 mg/dL (Dialysis patients were eligible) -Severe PVD	-Severe aorto-iliac arterial disease -severe aortic valvular disease -life expectancy of <1 year -Prior study participation (within 30 days) or recent CABG (within 1 week)	-Shock > 24 h -Mechanical MI complications -Severe AV disease or MV -Already established MCS (Impella or VA-ECMO) -Life expectancy of <1 year -Other shock causes (hypovolemia, sepsis, pulmonary embolism, or anaphylaxis) -Other conditions (Severe PAD; LV thrombus; IE; RV failure; OHCA with persistent GCS < 8; HIT)
Inclusion Criteria	-AMI within 48 h complicated by CS	-Age ≥ 18 -nonemergent PCI (ULM or last patent vessel with a LVEF ≤ 35%) -3xVessel disease with LVEF < 30%	-STEMI-CS undergoing immediate PCI -Mechanically ventilated before randomization	-Age ≥ 18 years -STEMI-CS
Impella Use	After PCI	During PCI	Before/immediately after PCI	Before/after PCI
Control	IABP (post PCI)	IABP (during PCI)	IABP (before or immediately after PCI)	Standard care
Key Endpoints	-CI at 30 min post-implantation -30 d all-cause mortality	-30 and 90 d major adverse events (all-cause mortality, stroke, repeat revascularization)	-30 d and 6 mo all-cause mortality	-180 d all-cause mortality -Need for additional MCS or heart transplant
Main Findings	-↑ CI, -No 30 d mortality difference	-No 3 d difference -Trend toward better 90 d outcomes	-No difference in 30 d or 6 mo all-cause mortality	-↓ 180 d all-cause mortality in Impella group
SCAI Stage	Primarily stage C	N/A	Primarily stage C	Primarily stage E
STEMI (%)	N/A	No	100%	100%
Lactate (mmol/L)	6.2	N/A	7.5 ± 3.2 (Impella) and 8.9 ± 6.6 (IABP)	4.5 (3.3–7.1)
MCS Timing	100% post-revascularization	100% pre-revascularization	80% (Impella) and 88% (IABP) post-revascularization	46.9% pre-revascularization
PAC Use	N/A	No	N/A	68%
Crossover	1 patient in mAFP arm did not receive it	N/A	Crossover or upgrading 4.2% mAFP and 12.5% IABP	1.7% crossed over to mAFP
Escalation	N/A	N/A	1 bridged to surgical LVAD in IABP arm	5.6% (Impella) and 2.3% (standard of care) bridged to LVAD

HOCM: Hypertrophic obstructive cardiomyopathy, PE: pulmonary embolism, AV: aortic valve, PVD: Peripheral vascular disease, OHCA: Out-of-hospital CA, GCS: Glascow coma scale, HIT: heparin-induced thrombocytopenia, ULM: Unprotected left main, CI: Cardiac index; MV: mechanical valve; ↑: Increased; ↓: Decreased. N/A: Not applicable as patients with cardiogenic shock were excluded from PROTECT II.

## Data Availability

No new data were created or analyzed in this study. Data sharing is not applicable to this article.

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
