# Peer review of "The Role of Impella in Cardiogenic Shock in the Post-DanGer Shock Era"

_biomedicines, 2025, doi:10.3390/biomedicines13092198_

Round 1

Reviewer 1 Report

Comments and Suggestions for Authors

This is a well written review of the clinical use and trial data of the Impella devices and highlights well the strengths and deficiencies of the present clinical data. It provides a good overview of the haemodynamic changes in cardiogenic shock and how Impella modifies these in comparison to IABP and ECHMO devices. I would like the authors to touch on the cost and cost effectiveness if possible, as this will be one reason why national societies and reimbursement authorities are being cautious.

Author Response

Reviewer 1

Comment:

“This is a well written review of the clinical use and trial data of the Impella devices and highlights well the strengths and deficiencies of the present clinical data. It provides a good overview of the hemodynamic changes in cardiogenic shock and how Impella modifies these in comparison to IABP and ECHMO devices. I would like the authors to touch on the cost and cost effectiveness if possible, as this will be one reason why national societies and reimbursement authorities are being cautious.”

Response:

We sincerely appreciate the reviewer’s thoughtful comments regarding the cost-effectiveness of Impella in the studied population. In response, we have updated the manuscript (page 12, lines 217–226) to include the following:

“It is also important to note that in addition to the limited supporting clinical data, substantial hospitalization costs have restricted the widespread adoption of the device 1,2.  The costs of mAFP use exceeded those of patients who did not receive MCS, reflecting not only the direct expense of the device itself but also the additional costs related to concomitant interventions and post-procedural care 3. These expenses were also evident when compared with IABP use. In an observational study by Khera et al., the mean hospitalization cost for mAFP recipients between 2007 and 2012 was estimated at $85,580 per patient, translating into a total cost of approximately $1 billion over that period, compared with a significantly lower mean cost of $55,168 for those receiving IABP 4. Evidence regarding the cost-effectiveness of Impella is mixed and remains controversial, with industry-sponsored studies often reporting favorable outcomes, while independent evaluations generally show higher costs without clear benefit 5. Updated real-world studies from the past decade are needed to better assess its cost-effectiveness and clinical impact.”

Reviewer 2 Report

Comments and Suggestions for Authors

In this insightful and well-structured review, the authors summarize the current body of evidence on the use of microaxial flow pumps (mAFP) in the setting of cardiogenic shock (CS). The article begins with a comprehensive description of the device, including its different models and their hemodynamic effects. It then reviews both observational and randomized studies evaluating its use, with particular focus on the most recent and pivotal trial—the DanGer Shock trial, which represents the first randomized evidence supporting early mAFP use in patients with acute myocardial infarction-related cardiogenic shock (AMI-CS). The review continues with an overview of guideline recommendations across different countries and concludes with an in-depth analysis of adverse events and device-related complications.

In my opinion, this is an excellent and thorough piece of work, offering a comprehensive and up-to-date overview of the current knowledge surrounding mAFP use. The authors have successfully highlighted both the potential benefits of this device and the unanswered questions that still need to be addressed, while also outlining future directions for clinical research.
Personally, I believe that mechanical circulatory support devices such as mAFP are critical in improving survival in this high-risk population. They play a key role in preserving end-organ perfusion and in left ventricular unloading during myocardial recovery. The current lack of robust evidence is, in my view, primarily a consequence of the clinical severity and high early mortality of the target population. For this reason, I strongly support the use of strict inclusion criteria—as done in the DanGer Shock trial—to select appropriate patients, even though this may limit the generalizability of the results to the broader AMI-CS population. Moreover, I fully agree with the authors on the importance of closely monitoring and minimizing device-related complications (e.g., vascular injury, thrombosis, and hemolysis), as these adverse events significantly impact patient outcomes and may reduce the adoption of a technology that otherwise holds considerable promise in the management of cardiogenic shock.

Author Response

Reviewer 2

Comment:

“In this insightful and well-structured review, the authors summarize the current body of evidence on the use of microaxial flow pumps (mAFP) in the setting of cardiogenic shock (CS). The article begins with a comprehensive description of the device, including its different models and their hemodynamic effects. It then reviews both observational and randomized studies evaluating its use, with particular focus on the most recent and pivotal trial—the DanGer Shock trial, which represents the first randomized evidence supporting early mAFP use in patients with acute myocardial infarction-related cardiogenic shock (AMI-CS). The review continues with an overview of guideline recommendations across different countries and concludes with an in-depth analysis of adverse events and device-related complications.

In my opinion, this is an excellent and thorough piece of work, offering a comprehensive and up-to-date overview of the current knowledge surrounding mAFP use. The authors have successfully highlighted both the potential benefits of this device and the unanswered questions that still need to be addressed, while also outlining future directions for clinical research.
Personally, I believe that mechanical circulatory support devices such as mAFP are critical in improving survival in this high-risk population. They play a key role in preserving end-organ perfusion and in left ventricular unloading during myocardial recovery. The current lack of robust evidence is, in my view, primarily a consequence of the clinical severity and high early mortality of the target population. For this reason, I strongly support the use of strict inclusion criteria—as done in the DanGer Shock trial—to select appropriate patients, even though this may limit the generalizability of the results to the broader AMI-CS population. Moreover, I fully agree with the authors on the importance of closely monitoring and minimizing device-related complications (e.g., vascular injury, thrombosis, and hemolysis), as these adverse events significantly impact patient outcomes and may reduce the adoption of a technology that otherwise holds considerable promise in the management of cardiogenic shock.”

Response:

We sincerely thank the reviewer for the thoughtful and encouraging comments on our manuscript. We greatly appreciate the recognition of the structured overview of current evidence on mAFP in cardiogenic shock, as well as the emphasis on the clinical significance of these devices in improving outcomes for this critically ill population.

We also value the reviewer’s perspective on the importance of strict inclusion criteria, such as those used in the DanGer Shock trial, in balancing the need for rigorous patient selection against broader generalizability. We fully agree that careful patient selection, together with close monitoring and minimization of device-related complications, will be crucial for translating the promise of Impella into meaningful improvements in survival.

The reviewer’s insightful remarks further reinforce the relevance of our review in highlighting both the current strengths of the evidence base and the key unanswered questions that remain to be addressed in future trials.

Reviewer 3 Report

Comments and Suggestions for Authors

Dear editor

I reviewed the article entitled “The Role of Impella in Cardiogenic Shock in the Post-DanGer Shock Era”

I congrats the authors for their effort. The article is well written and easy to follow. I have only some minor comments.

1- A technical fand descriptive figure of the mAFP devices should be included in the text.

2- A figure may be added the text: a patients who inserted  a mAFP device and showing the procedural details.

Sincerely.

Author Response

Reviewer 3

Comment:

“I reviewed the article entitled “The Role of Impella in Cardiogenic Shock in the Post-DanGer Shock Era”

I congrats the authors for their effort. The article is well written and easy to follow. I have only some minor comments.

1- A technical fand descriptive figure of the mAFP devices should be included in the text.

2- A figure may be added the text: a patients who inserted  a mAFP device and showing the procedural details.”

Response:

We sincerely thank the reviewer for their thoughtful and constructive comments.

  1. We appreciate the suggestion to include a technical and descriptive figure of the mAFP devices. We would like to clarify that Figure 2 in the current manuscript already serves this purpose, providing a technical overview and descriptive details of the three most commonly used mAFP devices.
  2. With regard to including a figure depicting procedural details of device insertion, we respectfully note that such an illustration would require a series of multiple (6–10) images to adequately capture the full process. Furthermore, the procedural steps of mAFP insertion may vary slightly according to manufacturer specifications. For this reason, and to avoid potential issues with liability, we believe that detailed procedural instruction is best deferred to the manufacturer’s training programs. Instead, our manuscript is designed to focus on the evidence, outcomes, complications, and clinical considerations surrounding Impella use, which is information most relevant to the practicing clinician.

Therefore, we feel that no additional figures are necessary for this manuscript, and we will maintain our focus on figures most relevant to clinical and investigational aspects.